# The evolutionary origin of bilaterian smooth and striated myocytes

**Thibaut Brunet\*[†], Antje HL Fischer[‡], Patrick RH Steinmetz[§], Antonella Lauri[¶], Paola Bertucci, Detlev Arendt\***

Developmental Biology Unit, European Molecular Biology Laboratory, Heidelberg, Germany

**Abstract** The dichotomy between smooth and striated myocytes is fundamental for bilaterian musculature, but its evolutionary origin is unsolved. In particular, interrelationships of visceral smooth muscles remain unclear. Absent in fly and nematode, they have not yet been characterized molecularly outside vertebrates. Here, we characterize expression profile, ultrastructure, contractility and innervation of the musculature in the marine annelid *Platynereis dumerilii* and identify smooth muscles around the midgut, hindgut and heart that resemble their vertebrate counterparts in molecular fingerprint, contraction speed and nervous control. Our data suggest that both visceral smooth and somatic striated myocytes were present in the protostome-deuterostome ancestor and that smooth myocytes later co-opted the striated contractile module repeatedly – for example, in vertebrate heart evolution. During these smooth-to-striated myocyte conversions, the core regulatory complex of transcription factors conveying myocyte identity remained unchanged, reflecting a general principle in cell type evolution.

**\*For correspondence:** t.brunet@ berkeley.edu (TB); arendt@embl. de (DA)

**Present address:** [†]Department of Molecular and Cell Biology, Howard Hughes Medical Institute, University of California, Berkeley, Berkeley, United States; [‡]Ludwig-Maximilians University Munich, Munich, Germany; [§]Sars International Centre for Marine Molecular Biology, University of Bergen, Bergen, Norway; [¶]Institute for Biological and Medical Imaging, Helmholtz Zentrum München, Neuherberg, Germany

**Competing interests:** The authors declare that no competing interests exist.

## Introduction

Musculature is composed of myocytes that are specialized for active contraction (*Schmidt-Rhaesa, 2007*). Their contractile apparatus centers on actomyosin, a contractile module that dates back to stem eukaryotes (*Brunet and Arendt, 2016a*) and incorporated accessory proteins of pre-metazoan origin (*Steinmetz et al., 2012*). Two fundamentally distinct types of myocytes are distinguished based on ultrastructural appearance. In striated myocytes, actomyosin myofibrils are organized in aligned repeated units (sarcomeres) separated by transverse 'Z discs', while in smooth myocytes adjacent myofibrils show no clear alignment and are separated by scattered 'dense bodies' (*Figure 1A*). In vertebrates, striated myocytes are found in voluntary skeletal muscles, but also at the anterior and posterior extremities of the digestive tract (anterior esophagus muscles and external anal sphincter), and in the muscular layer of the heart; smooth myocytes are found in involuntary visceral musculature that ensures slow, long-range deformation of internal organs. This includes the posterior esophagus and the rest of the gut, but also blood vessels, and most of the urogenital system. In stark contrast, in the fruit fly *Drosophila* virtually all muscles are striated, including gut visceral muscles (*Anderson and Ellis, 1967*; *Goldstein and Burdette, 1971*; *Paniagua et al., 1996*); the only exception are little-characterized multinucleated smooth muscles around the testes (*Susic-Jung et al., 2012*). Also, in the nematode *Caenorhabditis*, somatic muscles are striated, while the short intestine and rectum visceral myocytes are only one sarcomere-long and thus hard to classify (*Corsi et al., 2000*; *White, 1988*).

The evolutionary origin of smooth versus striated myocytes in bilaterians accordingly remains unsolved. Ultrastructural studies have consistently documented the presence of striated somatic myocytes in virtually every bilaterian group (*Schmidt-Rhaesa, 2007*) and in line with this, the comparison of Z-disc proteins supports homology of striated myocytes across bilaterians

(*Steinmetz et al., 2012*). The origin of smooth myocyte types, however, is less clear. Given the absence of smooth muscles from fly and nematode, it has been proposed that visceral smooth myocytes represent a vertebrate novelty, which evolved independently from non-muscle cells in the vertebrate stem line (*Goodson and Spudich, 1993*; *OOta and Saitou, 1999*). However, smooth muscles are present in many other bilaterian groups, suggesting instead their possible presence in urbilaterians and secondary loss in arthropods and nematodes. Complicating the matter further, intermediate ultrastructures between smooth and striated myocytes have been reported, suggesting interconversions (reviewed in [*Schmidt-Rhaesa, 2007*]).

Besides ultrastructure, the comparative molecular characterization of cell types can be used to build cell type trees (*Arendt, 2003*, *2008*; *Musser and Wagner, 2015*; *Wagner, 2014*). Cell type identity is established via the activity of transcription factors acting as terminal selectors (*Hobert, 2016*) and forming 'core regulatory complexes' (CoRCs; [*Arendt et al., 2016*; *Wagner, 2014*]), which directly activate downstream effector genes. This is exemplified for vertebrate myocytes in *Figure 1B*. In all vertebrate myocytes, transcription factors of the Myocardin family (MASTR in skeletal muscles, Myocardin in smooth and cardiac muscles) directly activate effector genes encoding contractility proteins (*Figure 1B*) (*Creemers et al., 2006*; *Meadows et al., 2008*; *Wang and Olson, 2004*; *Wang et al., 2003*). They heterodimerize with MADS-domain factors of the Myocyte Enhancer Factor-2 (Mef2) (*Black and Olson, 1998*; *Blais et al., 2005*; *Molkentin et al., 1995*; *Wales et al., 2014*) and Serum Response Factor (SRF) families (*Carson et al., 1996*; *Nishida et al., 2002*). Other myogenic transcription factors are specific for different types of striated and smooth myocytes. Myogenic Regulatory Factors (MRF) family members, including MyoD and its paralogs Myf5, Myogenin and Mrf4/Myf6 (*Shi and Garry, 2006*), directly control contractility effector genes in skeletal (and esophageal) striated myocytes, cooperatively with Mef2 (*Blais et al., 2005*; *Molkentin et al., 1995*) – but are absent from smooth and cardiac muscles. In smooth and cardiac myocytes, this function is ensured by NK transcription factors (Nkx3.2/Bapx and Nkx2.5/Tinman, respectively), GATA4/5/6 and Fox transcription factors (FoxF1 and FoxC1, respectively), which bind to SRF and Mef2 to form CoRCs directly activating contractility effector genes (*Durocher et al., 1997*; *Hoggatt et al., 2013*; *Lee et al., 1998*; *Morin et al., 2000*; *Nishida et al., 2002*; *Phiel et al., 2001*) (*Figure 1B*).

Regarding effector proteins (*Figure 1B*) (*Kierszenbaum and Tres, 2015*), all myocytes express distinct isoforms of the myosin heavy chain: the striated myosin heavy chain *ST-MHC* (which duplicated into cardiac, fast skeletal and slow skeletal isoforms in vertebrates) and the smooth/non-muscle myosin heavy chain *SM-MHC* (which duplicated in vertebrates into smooth *myh10*, *myh11* and *myh14*, and non-muscle *myh9*) (*Steinmetz et al., 2012*). The different contraction speeds of smooth and striated muscles are due to the distinct kinetic properties of these molecular motors (*Bárány, 1967*). In both myocyte types, contraction occurs in response to calcium, but the responsive proteins differ (*Alberts et al., 2014*): the Troponin complex (composed of Troponin C, Troponin T and Troponin I) for striated muscles, Calponin and Caldesmon for smooth muscles. In both myocyte types, calcium also activates the Calmodulin/Myosin Light Chain Kinase pathway (*Kamm and Stull, 1985*; *Sweeney et al., 1993*). Striation itself is implemented by specific effectors, including the long elastic protein Titin (*Labeit and Kolmerer, 1995*) (which spans the entire sarcomere and gives it elasticity and resistance) and ZASP/LBD3 (Z-band Alternatively Spliced PDZ Motif/LIM-Binding Domain 3), which binds actin and stabilizes sarcomeres during contraction (*Au et al., 2004*; *Zhou et al., 2001*). The molecular study of *Drosophila* and *Caenorhabditis* striated myocytes revealed important commonalities with their vertebrate counterparts, including the Troponin complex (*Beall and Fyrberg, 1991*; *Fyrberg et al., 1990*, *1994*; *Marín et al., 2004*; *Myers et al., 1996*), and a conserved role for Titin (*Zhang et al., 2000*) and ZASP/LBD3 (*Katzemich et al., 2011*; *McKeown et al., 2006*) in the striated architecture.

Finally, smooth and striated myocytes also differ physiologically. All known striated myocyte types (apart from the myocardium) strictly depend on nervous stimulations for contraction, exerted by innervating motor neurons. In contrast, gut smooth myocytes are able to generate and propagate automatic (or 'myogenic') contraction waves responsible for digestive peristalsis in the absence of nervous inputs (*Faussone-Pellegrini and Thuneberg, 1999*; *Sanders et al., 2006*). These autonomous contraction waves are modulated by the autonomic nervous system (*Silverthorn, 2015*). Regarding overall contraction speed, striated myocytes have been measured to contract 10 to 100 times faster than their smooth counterparts (*Bárány, 1967*).

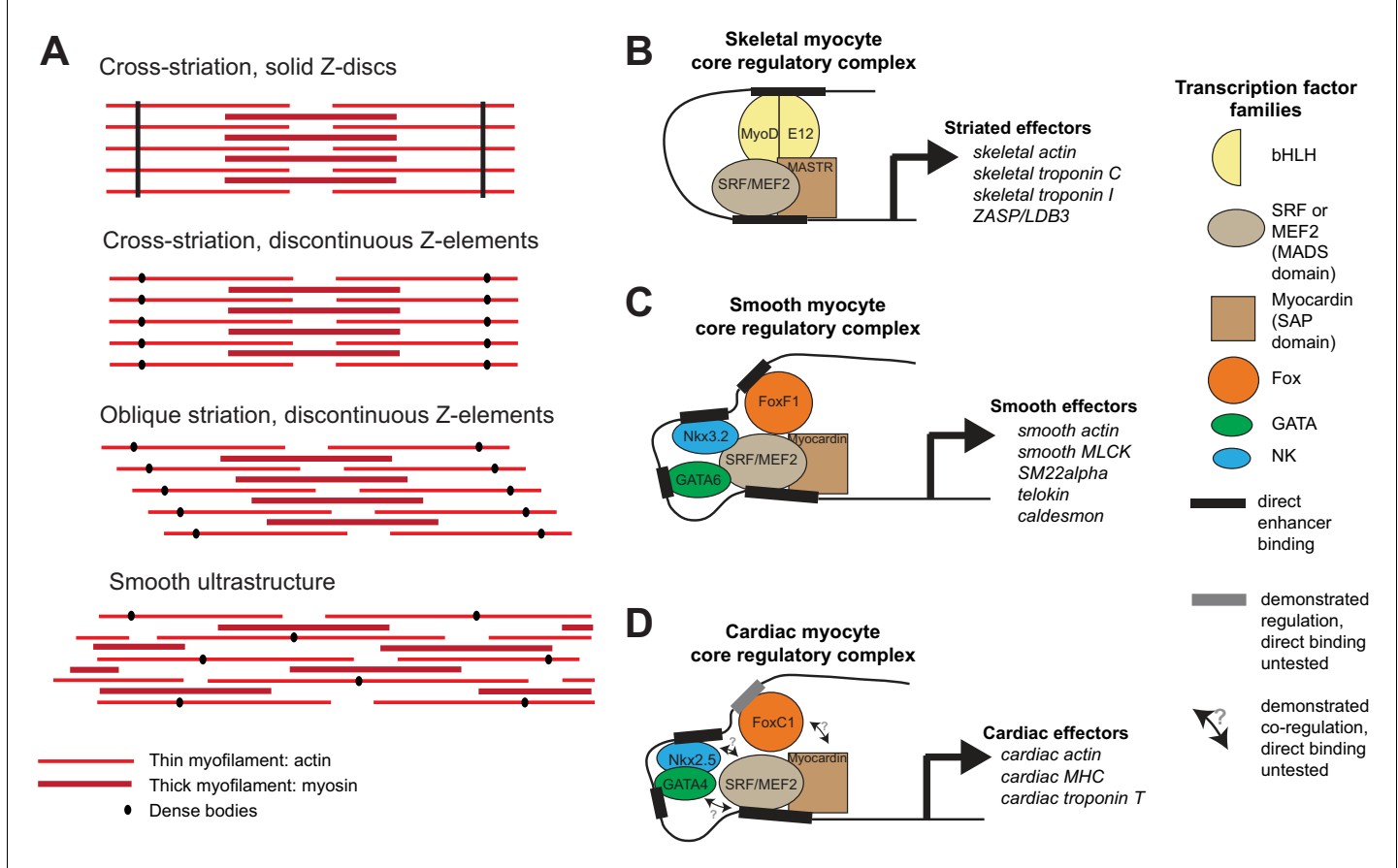

**Figure 1.** Ultrastructure and core regulatory complexes of myocyte types. (**A**) Schematic smooth and striated ultrastructures. Electron-dense granules called 'dense bodies' separate adjacent myofibrils. Dense bodies are scattered in smooth muscles, but aligned in striated muscles to form Z lines. (**B–D**) Core regulatory complexes (CoRCs) of transcription factors for the differentiation of different types of myocytes in vertebrates. Complexes composition from (*Creemers et al., 2006*; *Meadows et al., 2008*; *Molkentin et al., 1995*) for skeletal myocytes, (*Hoggatt et al., 2013*; *Nishida et al., 2002*; *Phiel et al., 2001*) for smooth myocytes and (*Durocher et al., 1997*; *Lee et al., 1998*) for cardiomyocytes. Target genes from (*Blais et al., 2005*) for skeletal myocytes, (*Nishida et al., 2002*) from smooth myocytes and (*Schlesinger et al., 2011*) for cardiomyocytes.

To elucidate the evolutionary origin and diversification of bilaterian smooth and striated myocytes, we provide an in-depth ultrastructural, molecular and functional characterization of the myocyte complement in the marine annelid *Platynereis dumerilii*, which belongs to the Lophotrochozoa. Strikingly, as of now, no invertebrate smooth visceral muscle has been investigated on a molecular level (*Hooper and Thuma, 2005*; *Hooper et al., 2008*). *Platynereis* has retained more ancestral features than flies or nematodes and is thus especially suited for long-range comparisons (*Denes et al., 2007*; *Raible et al., 2005*). Also, other annelids such as earthworms have been reported to possess both striated somatic and midgut smooth visceral myocytes based on electron microscopy (*Anderson and Ellis, 1967*). Our study reveals the parallel presence of smooth myocytes in the musculature of midgut, hindgut and pulsatile dorsal vessel and of striated myocytes in the somatic musculature and the foregut. *Platynereis* smooth and striated myocytes closely parallel their vertebrate counterparts in ultrastructure, molecular profile, contraction speed and reliance on nervous inputs, thus supporting the ancient existence of a smooth-striated duality in protostome/deuterostome ancestors.

## Results

### *Platynereis* midgut and hindgut muscles are smooth, while foregut and somatic muscles are striated

Differentiation of the *Platynereis* somatic musculature has been documented in much detail (*Fischer et al., 2010*) and, in 5 days post-fertilization (dpf) young worms, consists of ventral and dorsal longitudinal muscles, oblique and parapodial muscles, head muscles and the axochord (*Lauri et al., 2014*). At this stage, we found the first *Platynereis* visceral myocytes around the developing tripartite gut, which is subdivided into foregut, midgut and hindgut (based on the conserved regional expression of *foxA, brachyury* and *hnf4* gut specification factors [*Martín-Durán and Hejnol, 2015*]; *Figure 2—figure supplement 1*). At 7 dpf, visceral myocytes form circular myofibres around the foregut, and scattered longitudinal and circular fibres around midgut and hindgut (*Figure 2A*, *Figure 2—figure supplement 2A*), which expand by continuous addition of circular and longitudinal fibres to completely cover the dorsal midgut at 11dpf (*Figure 2A*, *Figure 2—figure supplement 2B*) and finally form a continuous muscular orthogon around the entire midgut and hindgut in the 1.5 months-old juvenile (*Figure 2A*, *Figure 2—figure supplement 2C*).

We then proceeded to characterize the ultrastructure of *Platynereis* visceral and somatic musculature by transmission electron microscopy (*Figure 2C–M*). All somatic muscles and anterior foregut muscles display prominent oblique striation with discontinuous Z-elements (*Figure 2C–H*; compare *Figure 1A*), as typical for protostomes (*Burr and Gans, 1998*; *Mill and Knapp, 1970*; *Rosenbluth, 1972*). To the contrary, visceral muscles of the posterior foregut, midgut and hindgut are smooth with scattered dense bodies (*Figure 2I–M*). The visceral muscular orthogon is partitioned into an external longitudinal layer and an internal circular layer (*Figure 2J*), as in vertebrates (*Marieb and Hoehn, 2015*) and arthropods (*Lee et al., 2006*). Thus, according to ultrastructural appearance, *Platynereis* has both somatic (and anterior foregut) striated muscles and visceral smooth muscles.

### The molecular profile of smooth and striated myocytes

We then set out to molecularly characterize annelid smooth and striated myocytes via a candidate gene approach. As a starting point, we investigated, in the *Platynereis* genome, the presence of regulatory and effector genes specific for smooth and/or striated myocytes in the vertebrates. We found striated muscle-specific and smooth muscle/non-muscle isoforms of both *myosin heavy chain* (consistently with published phylogenies [*Steinmetz et al., 2012*]) and *myosin regulatory light chain*. We also identified homologs of genes encoding calcium transducers (*calponin* for smooth muscles; *troponin I* and *troponin T* for striated muscles), striation structural proteins (*zasp/lbd3* and *titin*), and terminal selectors for the smooth (*foxF* and *gata456*) and striated phenotypes (*myoD*).

We investigated expression of these markers by whole-mount in situ hybridization (WMISH). Striated effectors are expressed in both somatic and foregut musculature (*Figure 3A,C*; *Figure 3—figure supplement 1*). Expression of all striated effectors was observed in every somatic myocyte group by confocal imaging with cellular resolution (*Figure 3—figure supplement 2*). Interestingly, *myoD* is exclusively expressed in longitudinal striated muscles, but not in other muscle groups (*Figure 3—figure supplement 2*).

The expression of smooth markers is first detectable at three dpf in a small triangle-shaped group of mesodermal cells posteriorly abutting the macromeres (which will form the future gut) (*Figure 3B*, *Figure 3—figure supplement 3A–C*). At this stage, smooth markers are also expressed in the foregut mesoderm (*Figure 3B*, *Figure 3—figure supplement 3A–C*, yellow arrows). At six dpf, expression of all smooth markers is maintained in the midgut and hindgut differentiating myocytes (*Figure 3D*, *Figure 3—figure supplement 3D–G*, *Figure 3—figure supplement 4A–E*) but smooth effectors disappear from the foregut, which turns on striated markers instead (*Figure 3—figure supplement 1R–W*) – reminiscent of the replacement of smooth fibres by striated fibres during development of the vertebrate anterior esophageal muscles (*Gopalakrishnan et al., 2015*). Finally, in 2-month-old juvenile worms, smooth markers are also detected in the dorsal pulsatile vessel (*Figure 3—figure supplement 3H–M*) – considered equivalent to the vertebrate heart (*Saudemont et al., 2008*) but, importantly, of smooth ultrastructure in polychaetes (*Jensen, 1974*; *Spies, 1973*). None of the striated markers is expressed around the midgut or the hindgut

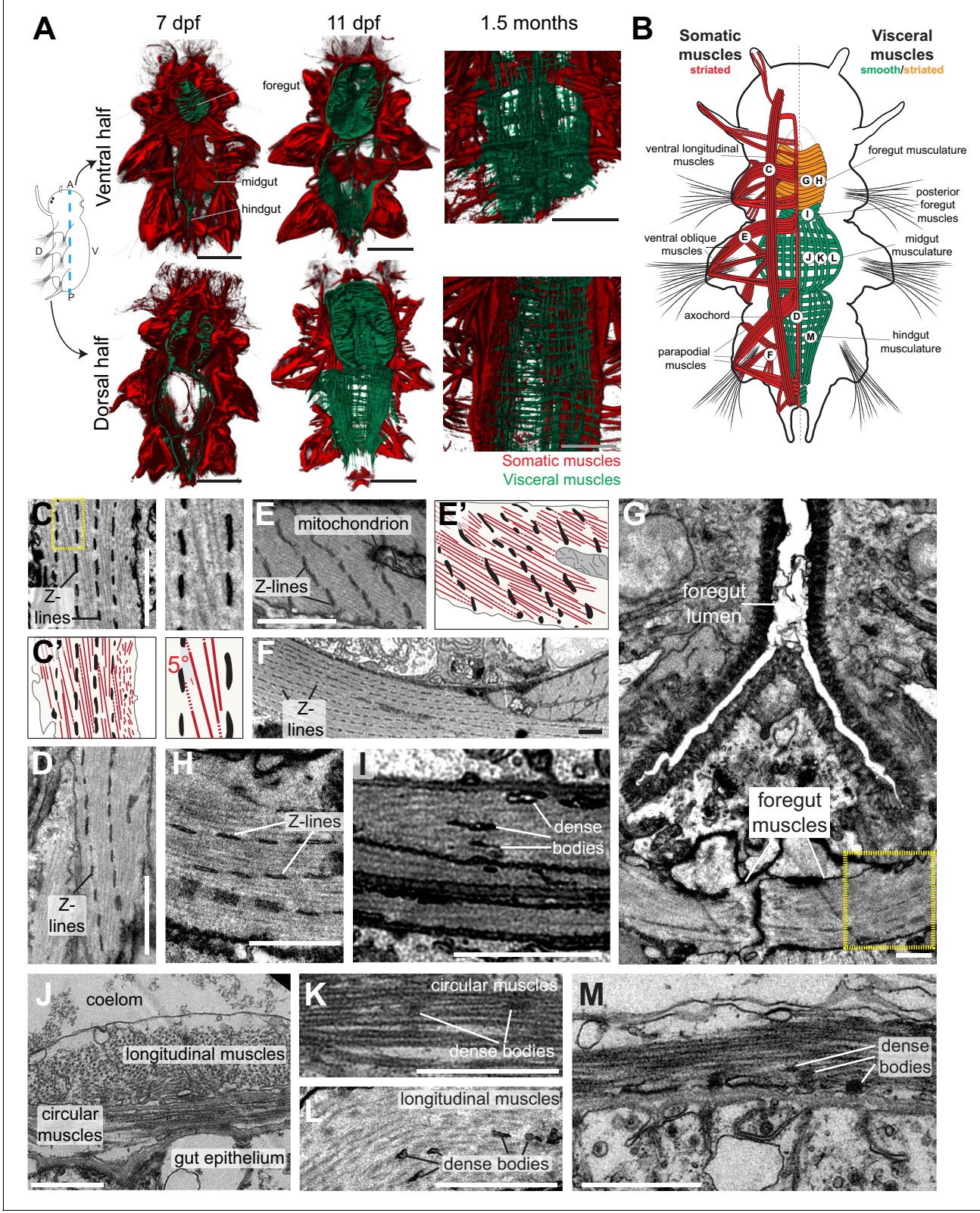

**Figure 2.** Development and ultrastructure of visceral and somatic musculature in *Platynereis* larvae and juveniles. (A) Development of visceral musculature. All panels are 3D renderings of rhodamine-phalloidin staining imaged by confocal microscopy. Visceral muscles have been manually colored green and somatic muscle red. Scale bar: 50 μm. (B) Schematic of the musculature of a late nectochaete (six dpf) larva. Body outline modified from (*Fischer et al., 2010*). Ventral view, anterior is up. (C–M) Electron micrographs of the main muscle groups depicted in B. Each muscle group is

*Figure 2 continued on next page*

*Figure 2 continued*

shown sectioned parallel to its long axis, so in the plane of its myofilaments. Scale bar: 2 μm. (C',E') are schematic drawings of the cells shown in (C,E). The Z-lines are made of aligned dense bodies (in black), myofilaments are in red, cytoplasm is in yellow and plasma membrane in grey. Attachment points of myofilaments on the dense bodies are represented with dotted lines when they are outside of the plane of section in the electron micrograph. Zoom panel in C' shows oblique striation with a 5° angle between myofilaments and Z-lines (compare to *Figure 1A*). (H) shows another cross-section in the stomodeum of the individual shown in G, in the region encased by the yellow box, and observed at a higher magnification. (J) shows the dorsal midgut in cross-section, dorsal side up.

The following figure supplements are available for figure 2:

**Figure supplement 1.** Gut patterning in *Platynereis* six dpf larvae.

**Figure supplement 2.** Formation of the visceral musculature observed in cross-section.

(*Figure 3—figure supplement 4F–K*), or in the dorsal vessel (*Figure 3—figure supplement 3L*). Taken together, these results strongly support conservation of the molecular fingerprint of both smooth and striated myocytes between annelids and vertebrates.

We finally investigated general muscle markers that are shared between smooth and striated muscles. These include *actin*, *mef2* and *myocardin* – which duplicated into muscle type-specific paralogs in vertebrates but are still present as single-copy genes in *Platynereis*. We found them to be expressed in the forming musculature throughout larval development (*Figure 3—figure supplement 5A–F*), and confocal imaging at six dpf confirmed expression of all three markers in both visceral (*Figure 3—figure supplement 5G–L*) and somatic muscles (*Figure 3—figure supplement 5M*).

## Smooth and striated muscles differ in contraction speed

We then characterized the contraction speed of the two myocyte types in *Platynereis* by measuring myofibre length before and after contraction, and by dividing the difference by the duration of contraction. Live confocal imaging of contractions in *Platynereis* larvae with fluorescently labeled musculature (*Video 1*, *Video 2*) gave a striated contraction rate of $0.55 \pm 0.27$ s$^{-1}$ (*Figure 4A–E*) and a smooth myocyte contraction rate of $0.07 \pm 0.05$ s$^{-1}$ (*Figure 4G*). As in vertebrates, annelid striated myocytes thus contract nearly one order of magnitude faster than smooth myocytes (*Figure 4F*).

## Striated, but not smooth, muscle contraction depends on nervous inputs

Finally, we investigated the nervous control of contraction of both types of muscle cells. In vertebrates, somatic muscle contraction is strictly dependent on neuronal inputs. By contrast, gut peristalsis is automatic (or myogenic – i.e. does not require nervous inputs) in vertebrates, cockroaches (*Nagai and Brown, 1969*), squids (*Wood, 1969*), snails (*Roach, 1968*), holothurians and sea urchins (*Prosser et al., 1965*). The only exceptions appear to be bivalves and malacostracans (crabs, lobster and crayfish), in which gut motility is neurogenic (*Prosser et al., 1965*). Regardless of the existence of an automatic component, the gut is usually innervated by nervous fibres modulating peristalsis movements (*Wood, 1969*; *Wu, 1939*).

Gut peristalsis takes place in *Platynereis* larvae and juveniles from six dpf onwards (*Video 3*), and we set out to test whether nervous inputs were necessary for it to take place. We treated 2-month-old juveniles with 180 μM Brefeldin A, an inhibitor of vesicular traffic which prevents polarized secretion (*Misumi et al., 1986*) and interferes with neurotransmission (*Malo et al., 2000*). Treatment stopped locomotion in all treated individuals, confirming that neurotransmitter release by motor neurons is required for somatic muscles contraction, while DMSO-treated controls were unaffected. On the other hand, vigorous gut peristalsis movements were maintained in Brefeldin-A-treated animals (*Video 4*). Quantification of the propagation speed of the peristalsis wave (*Figure 5A–D*; see Materials and methods) indicated that contractions propagated significantly faster in Brefeldin-A-treated individuals than in controls. The frequency of wave initiation and their recurrence (the number of repeated contraction waves occurring in one uninterrupted sequence) did not differ significantly in Brefeldin-A-treated animals (*Figure 5E,F*). These results indicate that, as in vertebrates,

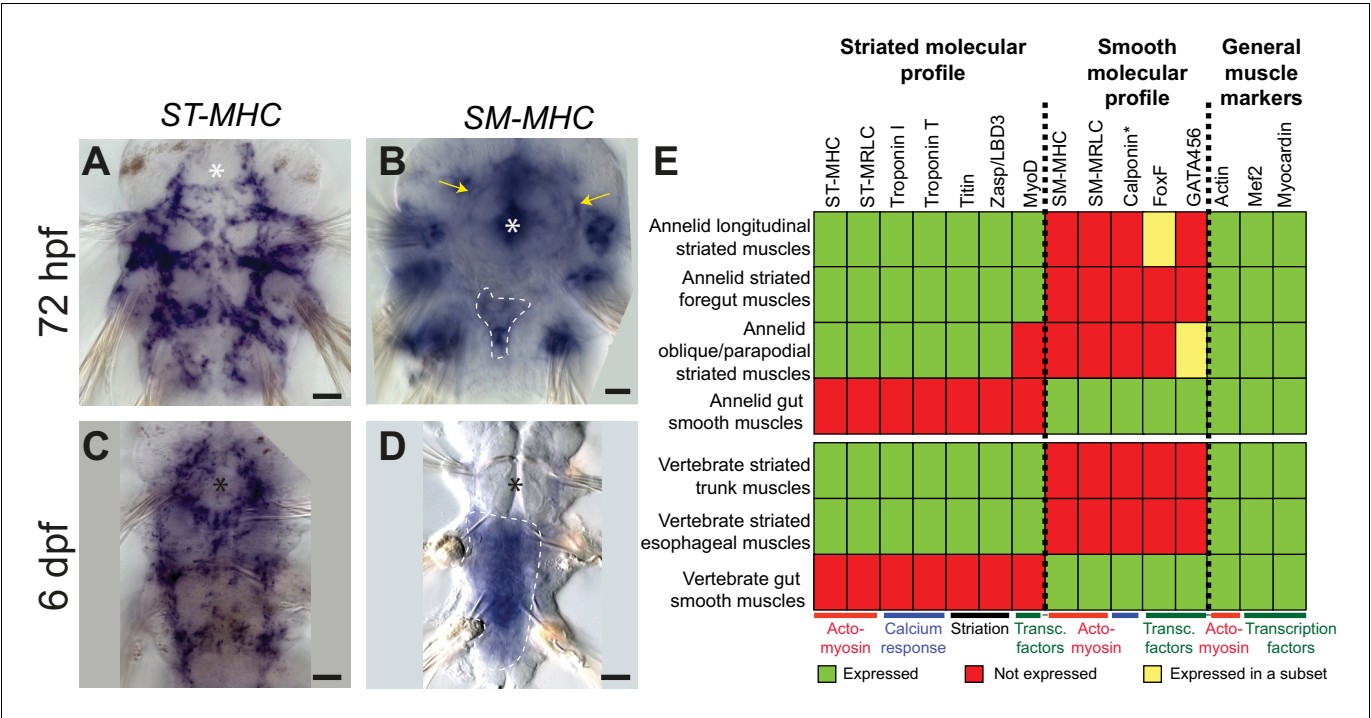

**Figure 3.** Expression of smooth and striated muscle markers in *Platynereis* larvae. Animals have been stained by WMISH and observed in bright-field Nomarski microscopy. Ventral views, anterior side up. Scale bar: 25 μm. (A–D) Expression patterns of the striated marker *ST-MHC* and the smooth marker *SM-MHC*. These expression patterns are representative of the entire striated and smooth effector module (see *Figure 3—figure supplement 1* and *Figure 3—figure supplement 3*). Note that *SM-MHC* (panel B) is expressed around the forming midgut and hindgut (dotted white line) as well as in the stomodeal sheath (white arrows) and in lateral cells in the parapodia. The identity of these cells is unknown, but preliminary observations suggest they will become part of the nephridial tubule/nephridiopore complex that opens at the base of the parapodia in annelids. Asterisk: stomodeum. (E) Table summarizing the expression patterns of smooth and striated markers in *Platynereis* and vertebrate muscles. (*) indicates that *Platynereis* and vertebrate Calponin are mutually most resembling by domain structure, but not one-to-one orthologs, as independent duplications in both lineages have given rise to more broadly expressed paralogs with a different domain structure (*Figure 7—figure supplement 3*).

The following figure supplements are available for figure 3:

**Figure supplement 1.** Expression of striated muscle markers in *Platynereis* larvae.

**Figure supplement 2.** Expression of striated muscle markers in the six dpf *Platynereis* larva.

**Figure supplement 3.** Expression of smooth muscle markers in *Platynereis* larvae.

**Figure supplement 4.** Molecular profile of midgut muscles in the six dpf larva.

**Figure supplement 5.** General muscle markers are expressed in both smooth and striated muscles.

visceral smooth muscle contraction and gut peristalsis do not require nervous (or secretory) inputs in *Platynereis*.

## An enteric nervous system is present in *Platynereis*

In vertebrates, peristaltic contraction waves are initiated by self-excitable myocytes (Interstitial Cajal Cells) and propagate across other smooth muscles by gap junctions ensuring direct electrical coupling (*Faussone-Pellegrini and Thuneberg, 1999*; *Sanders et al., 2006*). We tested the role of gap junctions in *Platynereis* gut peristalsis by treating animals with 2.5 mM 2-octanol, which inhibits gap junction function in both insects (*Bohrmann and Haas-Assenbaum, 1993*; *Gho, 1994*) and vertebrates (*Finkbeiner, 1992*). 2-Octanol abolishes gut peristalsis, both in the absence and in the

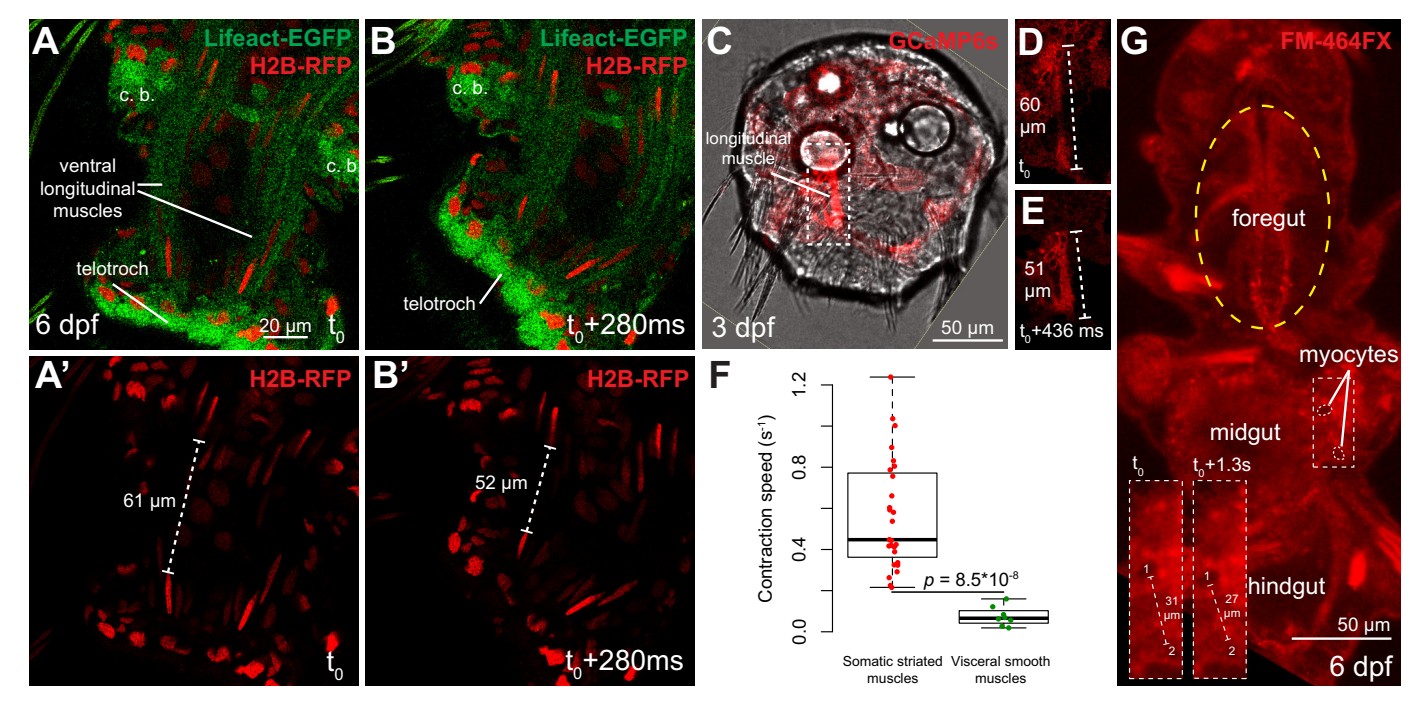

**Figure 4.** Contraction speed quantifications of smooth and striated muscles. (**A–B**) Snapshots of a time lapse live confocal imaging of a late nectochaete larva expressing fluorescent markers. Ventral view of the two posterior-most segments, anterior is up. (**C**) Snapshots of a time-lapse live confocal imaging of a three dpf larva expressing GCaMP6s. Dorsal view, anterior is up. (**D–E**) Two consecutive snapshots on the left dorsal longitudinal muscle of the larva shown in C, showing muscle contraction. (**F**) Quantification of smooth and striated muscle contraction speeds (see Experimental procedures and *Figure 4—source data 1*), p-value by Mann-Whitney's U test. Each point represents a biological replicate (see Materials and methods). (**G**) Snapshot of a time-lapse live confocal imaging of a late nectochaete larva. Ventral view, anterior is up. Optical longitudinal section at the midgut level.

The following source data is available for figure 4:

**Source data 1.** Contraction speed values measured for somatic and visceral muscles.

presence of Brefeldin A (*Figure 5G*), indicating that propagation of the peristalsis wave relies on direct coupling between smooth myocytes via gap junctions.

The acceleration of peristalsis upon Brefeldin A treatment indicates that gut peristalsis is modulated by secreted signals (neurotransmitters, hormones or neurohormones) whose net combined effect in normal, resting conditions is to slow down the self-generated peristaltic waves. This is consistent with the existence of neurotransmitters that inhibit visceral muscle contraction in other bilaterians such as vertebrates (adrenaline [*Burnstock, 1958*]) and squids (acetylcholine [*Wood, 1969*]). To gain insights into the nature of these secreted signals, we investigated the innervation of the *Platynereis* gut. Immunostainings of juvenile worms for acetylated tubulin revealed a dense, near-orthogonal nerve net around the entire gut (*Figure 6A*), which is tightly apposed to the visceral muscle layer (*Figure 6C*) and includes serotonergic neurites (*Figure 6B,C*) and cell bodies (*Figure 6D*), as well as previously described neurons expressing the conserved neuropeptide Myoinhibitory Peptide, that stimulates gut peristalsis (*Williams et al., 2015*). Interestingly, some enteric serotonergic cell bodies are devoid of neurites, thus resembling the vertebrate (non-neuronal) enterochromaffine cells – endocrine serotonergic cells residing around the gut and activating gut peristalsis by direct serotonin secretion upon mechanical stretch (*Bulbring and Crema, 1959*).

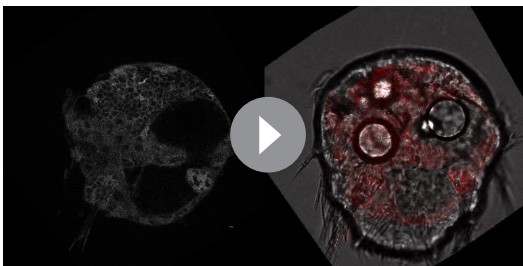

**Video 1.** Live imaging of somatic muscle contraction visualized by GCaMP6s. Dorsal view of a three dpf *Platynereis* larva injected (at the zygote stage) with a mRNA encoding GCaMP6s and mounted in 3% LMP agar between a slide and a cover slip. Anterior side is up. Left side is the red (GCaMP6s) fluorescence channel, right side shows overlay of transmitted light and red fluorescence channel. Time step between two frames: 0.436 s.

## Discussion

### Smooth and striated myocyte coexisted in bilaterian ancestors

Our study represents the first molecular characterization of protostome visceral smooth musculature (*Hooper and Thuma, 2005*; *Hooper et al., 2008*). The conservation of molecular signatures for both smooth and striated myocytes indicates that a dual musculature already existed in bilaterian ancestors: a fast striated somatic musculature (possibly also present around the foregut – as in *Platynereis*, vertebrates [*Gopalakrishnan et al., 2015*] and sea urchins [*Andrikou et al., 2013*; *Burke, 1981*]), under strict nervous control; and a slow smooth visceral musculature around the midgut and hindgut, able to undergo automatic peristalsis due to self-excitable myocytes directly coupled by gap junctions. In striated myocytes, a core regulatory complex (CoRC) involving Mef2 and Myocardin directly activated striated contractile effector genes such as *ST-MHC, ST-MRLC* and the *Troponin* genes (*Figure 7—figure supplement 1*). Notably, *myoD* might have been part of the CoRC in only part of the striated myocytes, as it is only detected in longitudinal muscles in *Platynereis*. The absence of *myoD* expression in other annelid muscle groups is in line with the 'chordate bottleneck' concept (*Thor and Thomas, 2002*), according to which specialization for undulatory swimming during early chordate evolution would have fostered exclusive reliance on trunk longitudinal muscles, and loss of other (*myoD*-negative) muscle types. In smooth myocytes, a CoRC composed of NK3, FoxF and GATA4/5/6 together with Mef2 and Myocardin activated the smooth contractile effectors *SM-MHC, SM-MRLC* and *calponin* (*Figure 7—figure supplement 1*). In spite of their absence in flies and nematodes, gut myocytes of smooth ultrastructure are widespread in other bilaterians, and an ancestral state reconstruction retrieves them as present in the last common protostome/deuterostome ancestor with high

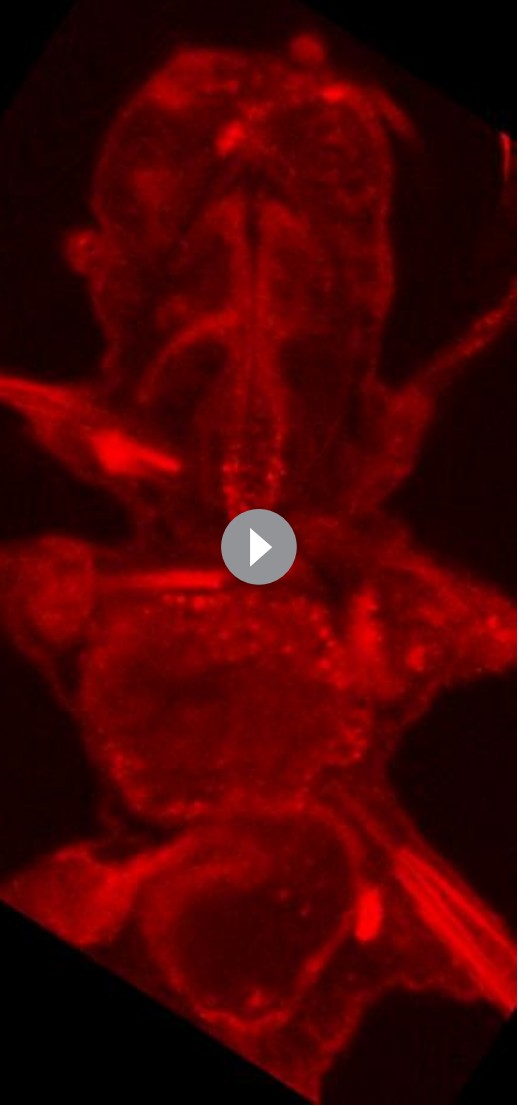

**Video 2.** Live imaging of visceral muscle contraction visualized by FM-464FX. Ventral view of a six dpf *Platynereis* larva stained with the vital dye FM-464FX and mounted in 3% LMP agar between a slide and a cover slip. Red fluorescence signal is shown. Anterior side is up. Time step between two frames: 1.29 s.

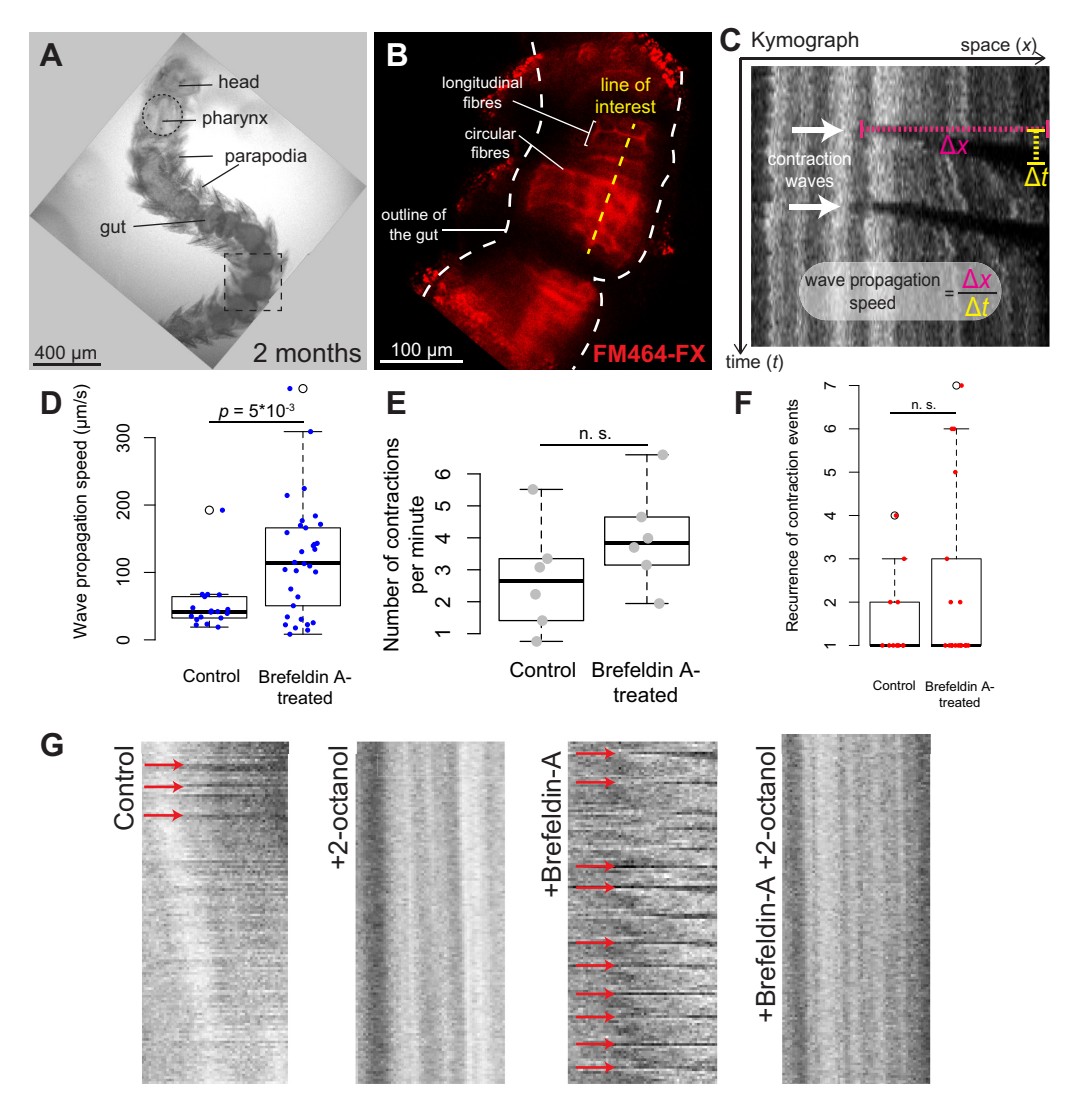

**Figure 5.** *Platynereis* gut peristalsis is independent of nervous inputs and dependent on gap junctions. (**A**) Two-month-old juvenile mounted in 3% low-melting point (LMP) agarose for live imaging. (**B**) Snapshots of a confocal live time-lapse imaging of the animal shown in **A**. Gut is observed by detecting fluorescence of the vital membrane dye FM-464FX. (**C**) Kymograph of gut peristalsis along the line of interest in (**B**). Contraction waves appear as dark stripes. A series of consecutive contraction waves is called a *contraction event*: here, two contraction waves are visible, which make up one contraction event with a recurrence of 2. (**D**) Quantification of the propagation speed of peristaltic contraction waves in mock (DMSO)-treated individuals and Brefeldin-A-treated individuals (inhibiting neurotransmission). Speed is calculated from kymographs (see Materials and methods and *Figure 5—source data 1*), p-value by Mann-Whitney's U test. Each point represents a contraction wave. Five biological replicates for each category (see Materials and methods). (**E,F**) Same as in **E**, but showing respectively the frequency of initiation and the recurrence of contraction events. Each point represents a biological replicate (see Materials and methods). (**G**) Representative kymographs of controls, animals treated with Brefeldin A (inhibiting neurotransmission), animals treated with 2-octanol (inhibiting gap junctions) and animals treated with both (N = 10 for each condition). 2-Octanol entirely abolishes peristaltic waves with or without Brefeldin A.

The following source data is available for figure 5:

**Source data 1.** Peristalsis waves quantifications in control and Brefeldin A-treated worms.

confidence (*Figure 7—figure supplement 2*), supporting our homology hypothesis. Our results are consistent with previous reports of Calponin immunoreactivity in intestinal muscles of earthworms (*Royuela et al., 1997*) and snails (which also lack immunoreactivity for Troponin T) (*Royuela et al., 2000*).

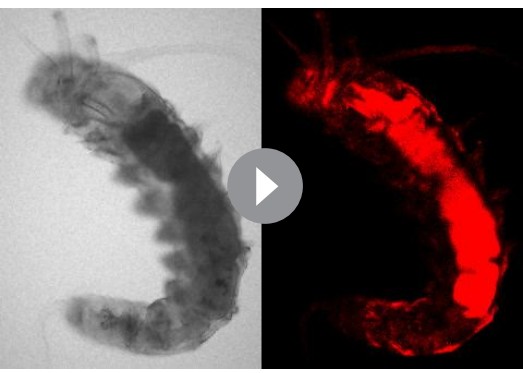

**Video 3.** Live imaging of gut peristalsis in a control 2-month-old juvenile worm. Lateral view of an individual stained with FM-464FX and mounted in 3% LMP agar between a slide and a cover slip. Left side is the transmitted light signal and right side is the red fluorescence channel. Note the peristalsis waves travelling along the gut, interrupted with rest periods.

## Origin of the enteric nervous system and enterochromaffine cells

In both *Platynereis* and vertebrates, visceral smooth myocytes are able to contract automatically but undergo modulation by secretory cells that form an enteric nerve plexus. Interestingly, an enteric nervous system has been found in most bilaterians investigated, including *Platynereis* (this study), earthworms (*Barna et al., 2001*; *Csoknya et al., 1991*; *Telkes et al., 1996*), snails (*Furukawa et al., 2001*), insects (*Copenhaver and Taghert, 1989*), nematodes (*Brownlee et al., 1994*) and echinoderms (*García-Arrarás et al., 1991*, *2001*). This suggests that the urbilaterian ancestor already possessed enteric neurons. In vertebrates, the enteric nervous system is entirely produced by the neural crest (*Le Douarin and Teillet, 1973*), a specialized migratory embryonic lineage which is a vertebrate innovation (*Shimeld and Holland, 2000*). This suggests that the neural crest 'took over' the production of the pre-existing enteric neurons (as it did with pharyngeal cartilage, of endomesodermal origin in stem-chordates (*Meulemans and Bronner-Fraser, 2007*), but produced by the neural crest in amniotes [*Lièvre and Le Douarin, 1975*; *Sefton et al., 2015*]). Alternatively, the ancient enteric neurons could have been lost in stem-vertebrates and later replaced by a novel, neural-crest-derived population. A careful comparison of the molecular fingerprints of invertebrate and vertebrate enteric neurons will be required to distinguish between these competing hypotheses. Alongside the enteric nervous system

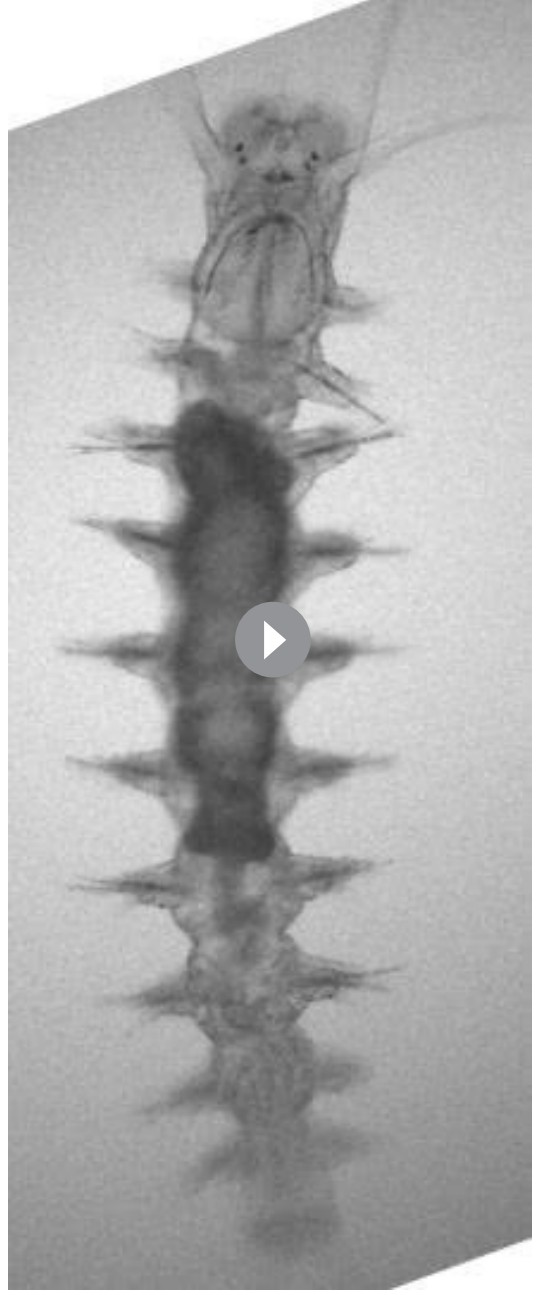

**Video 4.** Live imaging of gut peristalsis in a Brefeldin-A-treated 2-month-old juvenile worm. Ventral view of an individual treated with 180 µM Brefeldin-A, stained with FM-464FX (not shown) and mounted in 3% LMP agar between a slide and a cover slip. Transmitted light signal is shown. Note the vigorous and constant gut peristalsis waves travelling along the gut. The straight posture of the animal (compare with its bent control sibling in *Video 3*) is an effect of somatic muscle inhibition by Brefeldin-A.

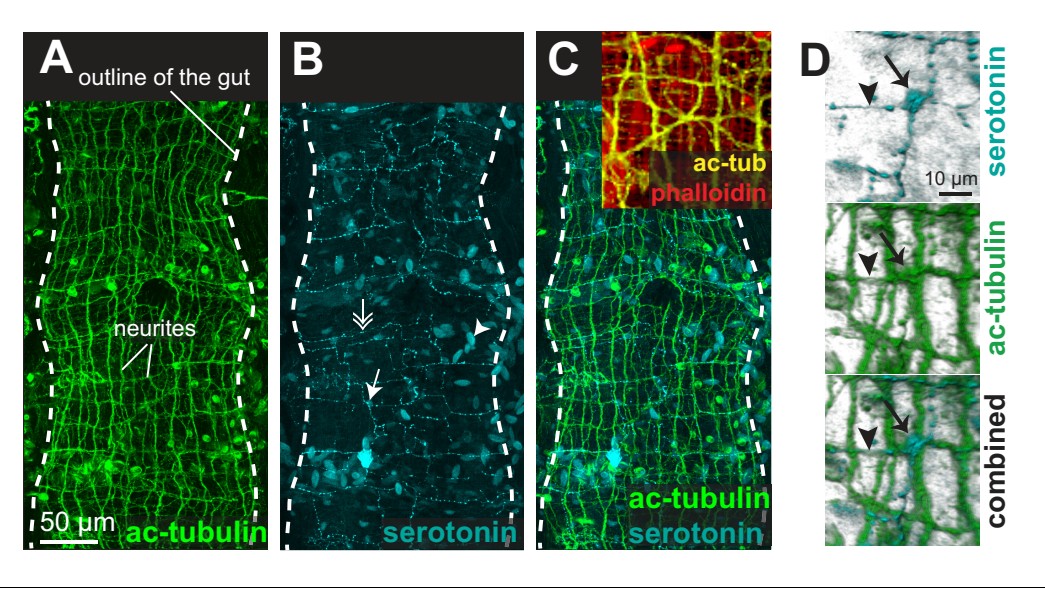

**Figure 6.** The enteric nerve net of *Platynereis*. (**A**) Immunostaining for acetylated tubulin, visualizing neurites of the enteric nerve plexus. Z-projection of a confocal stack at the level of the midgut. Anterior side up. (**B**) Same individual as in A, immunostaining for serotonin (5-HT). Note serotonergic neurites (double arrow), serotonergic neuronal cell bodies (arrow, see D), and serotonergic cell bodies without neurites (arrowhead). (**C**) Same individual as in A showing both acetylated tubulin and 5-HT immunostainings. Snapshot in the top right corner: same individual, showing both neurites (acetylated tubulin, yellow) and visceral myofibres (rhodamine-phalloidin, red). The acetylated tubulin appears yellow due to fluorescence leaking in the rhodamine channel. (**D**) 3D rendering of the serotonergic neuron shown by arrow in B.

(which includes serotonergic neurons in both vertebrates and annelids), the gut wall of both *Platynereis* and vertebrates also harbors non-neuronal, paracrine serotonergic cells (or enterochromaffine cells) – which are, unlike enteric neurons, of endodermal origin in vertebrates (*Andrew, 1974*; *Fontaine and Le Douarin, 1977*), and potentially represent another ancient bilaterian cell type modulating gut peristalsis.

## Origin of smooth and striated myocytes by cell type individuation

How did smooth and striated myocytes diverge in evolution? *Figure 7* presents a comprehensive cell type tree for the evolution of myocytes, with a focus on Bilateria. This tree illustrates the divergence of the two muscle cell types by progressive partitioning of genetic information in evolution – a process called *individuation* (*Arendt et al., 2016*; *Wagner, 2014*). The individuation of fast and slow contractile cells involved two complementary processes: (1) changes in CoRC (black circles, *Figure 7*) and (2) emergence of novel genes encoding new cellular modules, or *apomeres* (*Arendt et al., 2016*) (grey squares, *Figure 7*).

Around a common core formed by the Myocardin:Mef2 complex (both representing transcription factors of pre-metazoan ancestry [*Steinmetz et al., 2012*]), smooth and striated CoRCs incorporated different transcription factors implementing the expression of distinct effectors (*Figure 1B*; *Figure 7—figure supplement 1*) – notably the bilaterian-specific bHLH factor MyoD (*Steinmetz et al., 2012*) and GATA4/5/6, which arose by bilaterian-specific duplication of a single ancient pan-endo-mesodermal GATA transcription factor (*Leininger et al., 2014*; *Martindale et al., 2004*).

Regarding the evolution of myocyte-specific apomeres, one prominent mechanism of divergence has been gene duplication. While the *MHC* duplication predated metazoans, other smooth and striated-specific paralogs only diverged in bilaterians. Smooth and striated *MRLC* most likely arose by gene duplication in the bilaterian stem-line (*Supplementary file 1*). *Myosin essential light chain*, *actin* and *myocardin* paralogs split even later, in the vertebrate stem-line (*Figure 7*). Similarly, smooth and non-muscle *mhc* and *mrlc* paralogs only diverged in vertebrates. The *calponin*-encoding

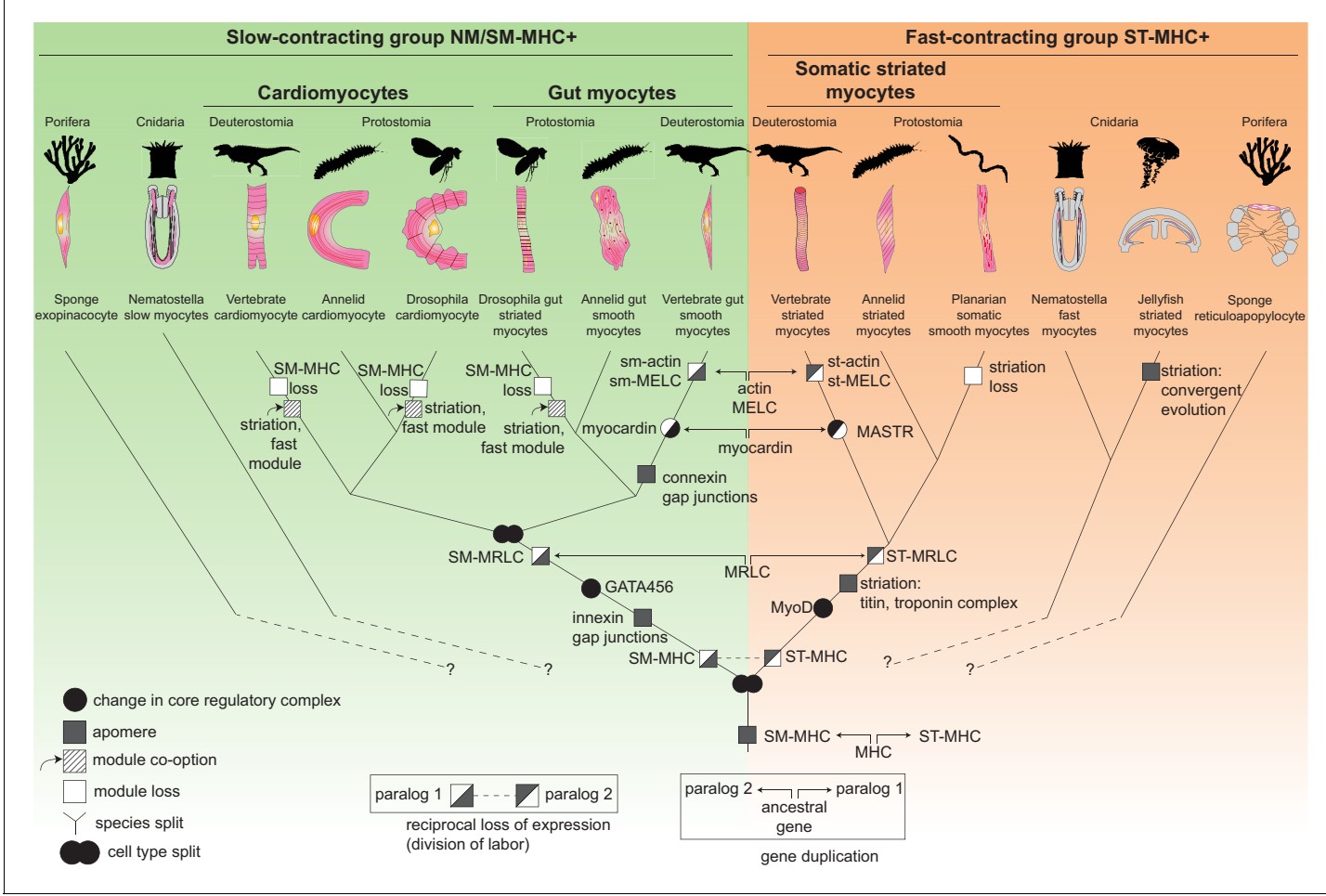

**Figure 7.** The evolutionary tree of animal contractile cell types. Bilaterian smooth and striated muscles split before the last common protostome/deuterostome ancestor. Bilaterian myocytes are split into two monophyletic cell type clades: an ancestrally *SM-MHC+* slow-contracting clade (green) and an ancestrally *ST-MHC+* fast-contracting clade (orange). Hypothetical relationships of the bilaterian myocytes to the *SM-MHC+* and *ST-MHC+* contractile cells of non-bilaterians are indicated by dotted lines (*Steinmetz et al., 2012*). Apomere: derived set of effector genes common to a monophyletic group of cell types (*Arendt et al., 2016*). Note that ultrastructure only partially reflects evolutionary relationships, as striation can evolve convergently (as in medusozoans), be co-opted (as in insect gut myocytes or in vertebrate and insect cardiomyocytes), be blurred, or be lost (as in planarians). Conversion of smooth to striated myocytes took place by co-option of striation proteins (Titin, Zasp/LDB3) and of the fast contractile module (ST-MHC, ST-MRLC, Troponin complex) in insect cardiomyocytes and gut myocytes, as well as in vertebrate cardiomyocytes. Nodes can either represent cell type duplications (indicated by two partly overlapping circles) or speciation events, as typical for a cell type tree (*Arendt, 2008*; *Serb and Oakley, 2005*).

The following figure supplements are available for figure 7:

**Figure supplement 1.** Evolution of myogenic Core Regulatory Complexes (CoRCs) in Bilateria.

**Figure supplement 2.** Ancestral state reconstructions of the ultrastructure of midgut/hindgut and heart myocytes.

**Figure supplement 3.** Domain structure, phylogeny and expression patterns of members of the *calponin* gene family.

gene underwent parallel duplication and subfunctionalization in both annelids and chordates, giving rise to both specialized smooth muscle paralogs and more broadly expressed copies with a different domain structure (*Figure 7—figure supplement 3*). This slow and stepwise nature of the individuation process is consistent with studies showing that recently evolved paralogs can acquire differential

expression between tissues that diverged long before in evolution (*Force et al., 1999*; *Lan and Pritchard, 2016*).

Complementing gene duplication, the evolution and selective expression of entirely new apomeres also supported individuation: for example, Titin and all components of the Troponin complex are bilaterian novelties (*Steinmetz et al., 2012*). In vertebrates, the new gene *caldesmon* was incorporated into the smooth contractile module (*Steinmetz et al., 2012*).

## Smooth to striated myocyte conversion

Strikingly, visceral smooth myocytes were previously assumed to be a vertebrate innovation, as they are absent in fruit flies and nematodes (two groups which are exceptions in this respect, at least from ultrastructural criteria [*Figure 7—figure supplement 2A*]). This view was seemingly supported by the fact that the vertebrate smooth and non-muscle myosin heavy chains (MHC) arose by vertebrate-specific duplication of a unique ancestral bilaterian gene, orthologous to *Drosophila* non-muscle MHC (*Goodson and Spudich, 1993*) – which, as our results suggest, rather reflects gradual individuation of pre-existing cell types (see above). Strikingly, the striated gut muscles of *Drosophila* resemble vertebrate and annelid smooth gut muscles by transcription factor expression (*nk3/bagpipe* [*Azpiazu and Frasch, 1993*], *foxF/biniou* [*Jakobsen et al., 2007*; *Zaffran et al., 2001*]), even though they express the fast/striated contractility module (*Fyrberg et al., 1994*, *1990*; *Marín et al., 2004*). If smooth gut muscles are ancestral for protostomes, as our results indicate, this suggests that the smooth contractile module was replaced by the fast/striated module in visceral myocytes during insect evolution. Interestingly, chromatin immunoprecipitation assays (*Jakobsen et al., 2007*) show that the conserved visceral transcription factors *foxF/biniou* and *nk3/bagpipe* do not directly control contractility genes in *Drosophila* gut muscles (which are downstream *mef2* instead), but establish the morphogenesis and innervation of the visceral muscles, and control non-contractile effectors such as gap junctions – which are the properties these muscles seem to have conserved from their smooth ancestors. The striated gut myocytes of insects would thus represent a case of co-option of an effector module from another cell type, which happened at an unknown time during ecdysozoan evolution (*Figure 7*; *Figure 7—figure supplement 1*).

Another likely example of co-option is the vertebrate heart: vertebrate cardiomyocytes are striated and express fast myosin and troponin, but resemble smooth myocytes by developmental origin (from the splanchnopleura), function (automatic contraction and coupling by gap junctions) and terminal selector profile (*Figure 1B*). These similarities suggest that cardiomyocytes might stem from smooth myocytes that likewise co-opted the fast/striation module. Indicative of this possible ancestral state, the *Platynereis* dorsal pulsatile vessel (considered homologous to the vertebrate heart based on comparative anatomy and shared expression of *NK4/tinman* [*Saudemont et al., 2008*]) expresses the smooth, but not the striated, myosin heavy chain (*Figure 3—figure supplement 3H–M*). An ancestral state reconstruction based on ultrastructural data further supports the notion that heart myocytes were smooth in the last common protostome/deuterostome ancestor, and independently acquired striation in at least five descendant lineages (*Figure 7—figure supplement 2B*) – usually in species with large body size and/or fast metabolism.

## Striated to smooth conversions

Smooth somatic muscles are occasionally found in bilaterians with slow or sessile lifestyles – for example in the snail foot (*Faccioni-Heuser et al., 1999*; *Rogers, 1969*), the ascidian siphon (*Meedel and Hastings, 1993*), and the sea cucumber body wall (*Kawaguti and Ikemoto, 1965*). As an extreme (and isolated) example, flatworms lost striated muscles altogether, and their body wall musculature is entirely smooth (*Rieger et al., 1991*). Interestingly, in all cases that have been molecularly characterized, smooth somatic muscles express the same fast contractility module as their striated counterparts, including ST-MHC and the Troponin complex – in ascidians (*Endo and Obinata, 1981*; *Obinata et al., 1983*), flatworms (*Kobayashi et al., 1998*; *Sulbarán et al., 2015*; *Witchley et al., 2013*), and the smooth myofibres of the bivalve catch muscle (*Nyitray et al., 1994*; *Ojima and Nishita, 1986*). (It is unknown whether these also express *zasp* and *titin* in spite of the lack of striation). This suggests that these are somatic muscles that have secondarily lost striation (in line with the sessile lifestyle of ascidians and bivalves, and with the complete loss of striated muscles in flatworms). Alternatively, they might represent remnants of ancestral smooth somatic fibres that

would have coexisted alongside striated somatic fibres in the last common protostome/deuterostome ancestor. Interestingly, the fast contractile module is also expressed in acoel body wall smooth muscles (*Chiodin et al., 2011*); since acoels belong to a clade that might have branched off before all other bilaterians (*Cannon et al., 2016*) (although a position within deuterostomes has also been envisioned [*Bourlat et al., 2003*, *2006*; *Philippe et al., 2011*]), these could represent fast-contracting myocytes that never evolved striation in the first place, similar to those found in cnidarians. In all cases, the fast contractility module appears to represent a consistent synexpression group (i.e. its components are reliably expressed together), and a stable molecular profile of all bilaterian somatic muscles, regardless of the presence of morphologically overt striation. This confirms the notion that, even in cases of ambiguous morphology or ultrastructure, the molecular fingerprint of cell types holds clue to their evolutionary affinities.

## Implications for cell type evolution

In the above, genetically well-documented cases of cell type conversion (smooth to striated conversion in insect visceral myocytes and vertebrate cardiomyocytes), cells kept their ancestral CoRC of terminal selector transcription factors, while changing the downstream effector modules. This supports the recent notion that CoRCs confer an abstract identity to cell types, which remains stable in spite of turnover in downstream effectors (*Wagner, 2014*) – just as *hox* genes impart conserved abstract identity to segments of vastly diverging morphologies (*Deutsch, 2005*). Tracking cell-type-specific CoRCs through animal phylogeny thus represents a powerful means to decipher the evolution of cell types.

## Pre-bilaterian origins

If the existence of fast-contracting striated and slow-contracting smooth myocytes predated bilaterians – when and how did these cell types first split in evolution? The first evolutionary event that paved the way for the diversification of the smooth and striated contractility modules was the duplication of the striated myosin heavy chain-encoding gene into the striated isoform *ST-MHC* and the smooth/non-muscle isoform *SM-MHC.* This duplication occurred in single-celled ancestors of animals, before the divergence of filastereans and choanoflagellates (*Steinmetz et al., 2012*). Consistently, both *sm-mhc* and *st-mhc* are present in the genome of the filasterean *Ministeria* (although *st-mhc* was lost in other single-celled holozoans) (*Sebé-Pedrós et al., 2014*). Interestingly, *st-mhc* and *sm-mhc* expression appears to be segregated into distinct cell types in sponges, cnidarians (*Steinmetz et al., 2012*) and ctenophores (*Dayraud et al., 2012*), suggesting that a cell type split between slow and fast contractile cells is a common feature across early-branching metazoans (*Figure 7*). Given the possibility of *MHC* isoform co-option (as outlined above), it is yet unclear whether this cell type split happened once or several times. The affinities of bilaterians and non-bilaterians contractile cells remain to be tested from data on the CoRCs establishing contractile cell types in non-bilaterians.

## Conclusions

Our results indicate that the split between visceral smooth myocytes and somatic striated myocytes is the result of a long individuation process, initiated before the last common protostome/deuterostome ancestor. Fast- and slow-contracting cells expressing distinct variants of myosin II heavy chain (*ST-MHC* versus *SM-MHC*) acquired increasingly contrasted molecular profiles in a gradual fashion – and this divergence process continues to this day in individual bilaterian phyla. Blurring this picture of divergence, co-option events have led to the occasional replacement of the slow contractile module by the fast one, leading to smooth-to-striated myocyte conversions. Our study showcases the power of molecular fingerprint comparisons centering on effector and selector genes to reconstruct cell type evolution (*Arendt, 2008*). In the bifurcating phylogenetic tree of animal cell types (*Liang et al., 2015*), it remains an open question how the two types of contractile cells relate to other cell types, such as neurons (*Mackie, 1970*) or cartilage (*Brunet and Arendt, 2016b*; *Lauri et al., 2014*; *Tarazona et al., 2016*).

## Materials and methods

### Immunostainings and in situ hybridizations

Immunostaining, rhodamine-phalloidin staining, and WMISH were performed according to previously published protocols (*Lauri et al., 2014*). Antibodies against acetylated tubulin and serotonin were purchased from Sigma Aldrich (RRID:AB_477585) and ImmunoStar (RRID:AB_572263), respectively. Rhodamine-phalloidin was purchased from ThermoFischer Scientific (RRID:AB_2572408). For all stainings not involving phalloidin, animals were mounted in 97% TDE/3% PTw for imaging following (*Asadulina et al., 2012*). Phalloidin-stained larvae were mounted in 1% DABCO/glycerol instead, as TDE was found to quickly disrupt phalloidin binding to F-actin. Confocal imaging of stained larvae was performed using a Leica SPE and a Leica SP8 microscope. Stacks were visualized and processed with ImageJ 1.49v (RRID:SCR_003070). 3D renderings were performed with Imaris 8.1 (RRID:SCR_007370). Bright-field Nomarski microscopy was performed on a Zeiss M1 microscope. Z-projections of Nomarski stacks were performed using Helicon Focus 6.7.1 (RRID:SCR_014462).

### Transmission electron microscopy

TEM was performed as previously published (*Lauri et al., 2014*).

### Pharmacological treatments

Brefeldin A was purchased from Sigma Aldrich (B7561) and dissolved in DMSO to a final concentration of 5 mg/mL. Animals were treated with 50 µg/mL Brefeldin A in 6-well plates filled with 5 mL filtered natural sea water (FNSW). Controls were treated with 1% DMSO (which is compatible with *Platynereis* development and survival without noticeable effect). Other neurotransmission inhibitors were found to be ineffective on *Platynereis* (as they elicited no impairment of locomotion): tetanus toxin (Sigma Aldrich T3194; 100 µg/mL stock in distilled water) up to 5 µg/mL; TTX (Latoxan, L8503; 1 mM stock) up to 10 µM; Myobloc (rimabotulinum toxin B; Solstice Neurosciences) up to 1%; saxitoxin 2 HCl (Sigma Aldrich NRCCRMSTXF) up to 1%; and neosaxitoxin HCl (Sigma Aldrich NRCCRMNEOC) up to 1%. (±)−2-Octanol was purchased from Sigma Aldrich and diluted to a final concentration of 2.5 mM (2 µL in 5 mL FNSW). (±)−2-Octanol treatment inhibited both locomotion and gut peristalsis, in line with the importance of gap junctions in motor neural circuits (*Kawano et al., 2011*; *Kiehn and Tresch, 2002*). No sample size was computed before the experiments. At least two technical replicates were performed for each assay, with at least five biological replicates per sample per technical replicate. A technical replicate is a batch of treated individuals (together with their control siblings), and a biological replicate is a treated (or control sibling) individual.

### Live imaging of contractions

Animals were mounted in 3% low melting point agarose in FNSW (2-Hydroxyethylagarose, Sigma Aldrich A9414) between a slide and a cover slip (using five layers of adhesive tape for spacing) and imaged with a Leica SP8 confocal microscope. Fluorescent labeling of musculature was achieved either by microinjection of mRNAs encoding *GCaMP6s*, *LifeAct-EGFP* or *H2B-RFP*, or by incubation in 3 µM 0.1% FM-464FX (ThermoFisher Scientific, F34653). Contraction speed was calculated as $(l2-l1)/(l1*t)$, where $l1$ is the initial length, $l2$ the length after contraction, and $t$ the duration of the contraction. Kymographs and wave speed quantifications were performed with the ImageJ Kymograph plugin: http://www.embl.de/eamnet/html/kymograph.html. No sample size was computed before the experiments. At least two technical replicates were performed for each assay, with at least two biological replicates per sample per technical replicate. A technical replicate is a batch of treated individuals (together with their control siblings), and a biological replicate is a treated (or control sibling) individual.

### Ancestral state reconstruction

Ancestral state reconstructions were performed with Mesquite 3.04 using the Maximum Likelihood and Parsimony methods.

## Cloning

The following primers were used for cloning Platynereis genes using a mixed stages Platynereis cDNA library (obtained from 1, 2, 3, 5, 6, 10 and 14-days old larvae) and either the HotStart Taq Polymerase from Qiagen or the Phusion polymerase from New England BioLabs (for GC-rich primers):

| Gene name | Forward primer | Reverse primer |
| --- | --- | --- |
| *foxF* | CCCAGTGTCTGCATCCTTGT | CATGGGCATTGAAGGGGAGT |
| *zasp* | CATACCAGCCATCCCGTCC | AAATCAGCGAACTCCAGCGT |
| *troponin T* | TTCTGCAGGGCGCAAAGTCA | CGCTGCTGTTCCTTGAAGCG |
| *SM-MRLC* | TGGTGTTTGCAGGGCGGTCA | GGTCCATACCGTTACGGAAGCTTTT |
| *calponin* | ACGTGCGGTTTACGATTGGA | GCTGGCTCCTTGGTTTGTTC |
| *transgelin1* | GCTGCCAAGGGAGCTGACGC | ACAAAGAGCTTGTACCACCTCACCC |
| *myocardin* | GACACCAGTCCGAAGCTTGA | CGTGGTAGTAGTCGTGGTCG |

The following genes were retrieved from an EST plasmid stock: *SM-MHC* (as two independent clones that gave identical expression patterns) and *ST-MRLC*. Gene orthology (Supplementary fils 1) was determined by phylogenetic analysis using MrBayes (RRID:SCR_012067) or PhyML (*Guindon et al., 2010*) run from http://www.atgc-montpellier.fr/phyml/ (RRID:SCR_014629).

Other genes were previously published: *actin* and *ST-MHC* (under the name *mhc1-4*) (*Lauri et al., 2014*) and *GATA456* (*Gillis et al., 2007*).

## Acknowledgements

We thank Kaia Achim for the *hnf4* plasmid, Pedro Machado (Electron Microscopy Core Facility, EMBL Heidelberg) for embedding and sectioning samples for TEM, and the Arendt lab for feedback on the project. The work was supported by the European Research Council 'Brain Evo-Devo' grant (TB, PB and DA), European Union's Seventh Framework Programme project 'Evolution of gene regulatory networks in animal development (EVONET)' [215781–2] (AL), the Zoonet EU-Marie Curie early training network [005624] (AF), the European Molecular Biology Laboratory (AL, AHLF, and PRHS) and the EMBL International Ph.D. Programme (TB, AHLF, PRHS, AL).

## Additional information

### Funding

| Funder | Grant reference number | Author |
| --- | --- | --- |
| European Research Council | Brain Evo-Devo | Thibaut Brunet Paola Bertucci Detlev Arendt |
| Seventh Framework Programme | EVONET | Antonella Lauri |
| European Union-Marie Curie Early Training Network | ZOONET | Antje HL Fischer |
| European Molecular Biology Laboratory | International PhD Program | Thibaut Brunet Antje HL Fischer Patrick RH Steinmetz Antonella Lauri |

The funders had no role in study design, data collection and interpretation, or the decision to submit the work for publication.

### Author contributions

TB, Conception and design, Acquisition of data, Analysis and interpretation of data, Drafting or revising the article; AHLF, PRHS, AL, Acquisition of data, Analysis and interpretation of data,

Contributed unpublished essential data or reagents; PB, Acquisition of data, Analysis and interpretation of data; DA, Conception and design, Analysis and interpretation of data, Drafting or revising the article

Author ORCIDs
Thibaut Brunet, http://orcid.org/0000-0002-1843-1613
Detlev Arendt, http://orcid.org/0000-0001-7833-050X

## Additional files

### Supplementary files

• Supplementary file 1. Phylogenetic trees of the markers investigated. (A) Simplified Maximum Likelihood (ML) tree for Myosin Regulatory Light Chain (full tree in panel M), rooted with Calmodulin, which shares an EF-hand calcium-binding domain with MRLC. (B) ML tree for FoxF, rooted with FoxQ1, the probable closest relative of the FoxF family (*Shimeld et al., 2010*). (C) MrBayes tree for bilaterian ZASP/LBD3, rooted with the cnidarian ortholog (*Steinmetz et al., 2012*). (D) ML tree for bilaterian Myosin Heavy Chain, rooted at the (pre-bilaterian) duplication between smooth and striated MHC (*Steinmetz et al., 2012*). (E) MrBayes tree for Mef2, rooted by the first splice isoform of the cnidarian ortholog (*Genikhovich and Technau, 2011*). (F) MrBayes tree for Titin, rooted at the protostome/deuterostome bifurcation (Titin is a bilaterian novelty). (G) MrBayes tree for Troponin T, rooted at the protostome/deuterostome bifurcation (Troponin T is a bilaterian novelty). (H) MrBayes tree for Troponin I, rooted by the Calponin/Transgelin family, which shares an EF-hand calcium-binding domain with Troponin I. (I) MrBayes tree for MyoD, rooted at the protostome/deuterostome bifurcation (MyoD is a bilaterian novelty). (J) MrBayes tree for Myocardin, rooted at the protostome/deuterostome bifurcation (the *Drosophila myocardin* ortholog is established [*Han et al., 2004*]). (K) Complete MRLC tree. Species names abbreviations: Pdu: *Platynereis dumerilii*; Xenla: *Xenopus laevis*; Mus: *Mus musculus*; Hsa: *Homo sapiens*; Dre: *Danio rerio*; Gga: *Gallus gallus*; Dme: *Drosophila melanogaster*; Cte: *Capitella teleta*; Patvu: *Patella vulgata*; Brafl: *Branchiostoma floridae*; Nve or Nemv: *Nematostella vectensis*; Acdi: *Acropora digitifera*; Expal: *Exaiptasia pallida*; Rat: *Rattus norvegicus*; Sko: *Saccoglossus kowalevskii*; Limu or Lpo: *Limulus polyphemus*; Trib or Trca: *Tribolium castaneum*; Daph: *Daphnia pulex*; Prcau: *Priapulus caudatus*; Cgi or Cgig: *Crassostrea gigas*; Ling or Linan: *Lingula anatina*; Hdiv: *Haliotis diversicolor*; Apcal or Aca: *Aplysia californica*; Spu: *Strongylocentrotus purpuratus*; Poli or Polis: *Polistes dominula*; Cin or Cint: *Ciona intestinalis*; Hro: *Helobdella robusta*; Bos: *Bos taurus*; Capsa: *Capsaspora owczarzaki*; Thtr: *Thecamonas trahens*; Lpo:; Bga: *Biomphalaria glabrata*; Cel: *Caenorhabditis elegans*; Tt: *Terebratalia transversa*; Octo: *Octopus vulgaris*; Sma: *Schmidtea mediterranea*; Bbe: *Branchiostoma belcheri*, Batden: *Batrachochytrium dendrobatidis*; Monve: *Mortierella verticillata*; Alloma: *Allomyces macrogynus*; Salpun: *Spizellomyces punctatus*; Mucor: *Mucor racemosus*; Lichco: *Lichtheimia corymbifera*; Ephmu: *Ephydatia muelleri*; Sycon: *Sycon ciliatum*; Amqu: *Amphimedon queenslandica*; Osc: *Oscarella lobularis*; Metse: *Metridium senile*; Pfu: *Pinctada fucata*; Rypa: *Riftia pachyptila*; Plma: *Placopecten magellanicus*; Air: *Argopecten irradians*; Scolop: *Scolopendra gigantea*; Artfra: *Artemia franciscana*; Bmor: *Bombyx mori*; Loa: *Loa loa*; Necator-am: *Necator americanus*; Trichi: *Trichinella spiralis*; Asc: *Ascaris lumbricoides*; Wuch: *Wucheria bancrofti*; Ancy: *Ancylostoma duodenale*; Callorinc: *Callorhinchus milii*; Dana: *Danaus plexippus*; Anop: *Anopheles gambiae*; Asty: *Astyanax mexicanus*; Oreo: *Oreochromis niloticus*; Icta: *Ictalurus punctatus*.

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
