## [Decision Letter]

Thank you for submitting your article "The evolutionary origin of bilaterian smooth and striated myocytes" for consideration by *eLife*. Your article has been reviewed by two peer reviewers, and the evaluation has been overseen by Alejandro Sánchez Alvarado as a Reviewing Editor and Fiona Watt as the Senior Editor. The following individual involved in the review of your submission has agreed to reveal his identity: Martin J Cohn (Reviewer #3).

The reviewers have discussed the reviews with one another and the Reviewing Editor has drafted this decision to help you prepare a revised submission.

Summary:

In this manuscript, the authors identify striated and smooth muscle in the annelid *Platynereis* and discovers their respective molecular profiles, from which the authors derive homology with the vertebrate counterparts, and thus infer a divergence of these two cell types before the protostome/deuterostome split. This is of great relevance because the two most widely used protostome model organisms, flies and nematodes, lack obvious homologs of vertebrate smooth muscle, and such lack has contributed to the view that smooth muscle originated in the vertebrate lineage. Overall, the design of the experiments reported is logical, and the hypothesis that smooth and striated muscle cell types were present in the common ancestor of bilateria is supported by multiple and rigorously obtained lines of evidence. Moreover, the conclusions and interpretations are justified by the clear and convincing data presented in the main and supplemental figures. As such, this elegant study will be of great interest to an unusually broad group of readers.

Essential revisions:

Before this manuscript can be considered further for publication in *eLife*, the authors need to address the following concerns:

1) Results, first paragraph and Figure 2: The introductory information on the somatic and visceral muscles of *Platynereis* a little sketchy, for an audience 99% of whom has no acquaintance with it. The text refers to Fisher et al. for background, but the paper does not appear to mention "visceral muscles" or "gut muscles". The current text signals the first visceral myocytes at 5 days, but it is not shown. At 7 days and 11 days, all that is shown are manually color-coded 3D renderings – which are not that easy to decipher except for the general fact that visceral muscles are central to more peripheral somatic ones, which makes sense, but is not very informative. Therefore, barring an oversight on our part, the first published embryological and anatomical data on gut muscles in *Platynereis* is represented in Figure 2 of the current manuscript as a highly stylized and processed image. Could we have more "original" data, for example, in the form of transverse sections, optical or real, which would show the spatial arrangements of gut versus somatic muscles. This could also include those "intrinsic muscles" that appear in Figure 3—figure supplement 3, which on a schematic seem to radiate from the gut to the walls of the animal (but not so clearly at all in the adjacent panels N or P). Also, the relationship with the tripartite organization shown in Figure 2—figure supplement 1 is not visually explicit (and as it stands that supplementary figure is not very useful).

2) Figure 2 panels C-I: Concerning discontinuous Z lines: for a non-specialist, it is not clear from the EM what is oblique relative to what (in some panels the Z lines seem virtually parallel to other striation, presumably the filaments) and what is the relationship with the schematics in Figure 1. Maybe a higher magnification would help? Contrary to what the legend says, H is not an enlargement of the yellow box in G (even if it looks similar).

3) Figure 3, panel B: smooth muscle MHC is expressed in many more places than the small posterior triangle at 72hpf: this should be commented on. Figure 3—figure supplement 3, panel J: Why is actin considered a muscle marker? Is it smooth muscle actin? The Panels J and K look like featureless clouds of fluorescent color that could easily be made to overlap more or less by adjusting the intensity and are not very convincing. In general, Figure 3—figure supplement 3 in its current form is probably the least satisfactory piece of expression data: expression of Gata456 in the gut region is not clear at 72hpf, (if anything, on the wholemount, more of a gap than a signal in the posterior midline is what is seen) and at 6dpf it is embedded in a ubiquitous-looking expression in the posterior half of the animal, which weakens the correlation. FoxF1 and NK3 are expressed in a dotted or very patchy pattern hard to relate to MRLC and calponin. On the schematic, the red and orange should be differentiated more. These ambiguities need to be resolved.

4) Figure 3—figure supplement 4: Why such different shapes of the midgut given that we are looking at Z stacks of confocal views: we would expect that a more homogenous morphology could be achieved from animal to animal. Without the dashed line it would be difficult to conclude that we are looking at the same structure.

---

## [Author Response]

*Essential revisions:*

*Before this manuscript can be considered further for publication in eLife, the authors need to address the following concerns:*

*1) Results, first paragraph and Figure 2: The introductory information on the somatic and visceral muscles of Platynereis a little sketchy, for an audience 99% of whom has no acquaintance with it. The text refers to Fisher et al. for background, but the paper does not appear to mention "visceral muscles" or "gut muscles". The current text signals the first visceral myocytes at 5 days, but it is not shown.*

The first elements of visceral musculature detectable by phalloidin stainings are indeed found at 5 days post-fertilization, as a few scattered and disconnected myofibres of minute size on the ventral body side. Figure 2—figure supplement 1 shows a picture of a slightly older individual (6 dpf), stained with phalloidin. The first elements of the incipient visceral musculature are isolated myofibres that can be recognized, in this panel and in the following, by its position immediately basal to the midgut epithelium (arrow).

*At 7 days and 11 days, all that is shown are manually color-coded 3D renderings – which are not that easy to decipher except for the general fact that visceral muscles are central to more peripheral somatic ones, which makes sense, but is not very informative. Therefore, barring an oversight on our part, the first published embryological and anatomical data on gut muscles in Platynereis is represented in Figure 2 of the current manuscript as a highly stylized and processed image. Could we have more "original" data, for example, in the form of transverse sections, optical or real, which would show the spatial arrangements of gut versus somatic muscles.*

We have now included a full additional supplementary figure (Figure 2—figure supplement 2) compiling virtual cross-sections of confocal stacks of the *Platynereis* musculature at 7 dpf, 11 dpf, and 1.5 months post-fertilization, as well as interpretative schematics. These new panels illustrate in detail how the visceral muscles come to surround the gut as a circular layer crossed by perpendicular transverse myofibres. As explained above, the visceral musculature can be recognized by its basal position directly adjacent to the midgut epithelium. In cases where it was not directly apparent, we found that the midgut epithelium could be made visible by enhancing the green (phalloidin) signal, and that information was used to draw its outline in the schematic drawings on the right side.

*This could also include those "intrinsic muscles" that appear in Figure 3—figure supplement 3, which on a schematic seem to radiate from the gut to the walls of the animal (but not so clearly at all in the adjacent panels N or P).*

The intrinsic muscles are visible and labeled in Figure 2—figure supplement 2 panels A-A’ and B-B’. These new data show that the intrinsic muscles cross the body cavity from dorsal to ventral and are located around the gut (more peripherally than the visceral musculature) rather than radiating from it as we had previously thought. We have corrected our interpretation accordingly in the schematic Figure 3—figure supplement 3. We thank the referee for raising this point.

*Also, the relationship with the tripartite organization shown in Figure 2—figure supplement 1 is not visually explicit (and as it stands that supplementary figure is not very useful).*

We have completed Figure 2—figure supplement 1 with ventral and lateral views of phalloidin stainings that make the tripartite gut organization more evident and help to interpret the gene expression panels. This scheme also illustrates the position of the sectional plane in Figure 2—figure supplement 2 (though the animal in Figure 2—figure supplement 1 is one day younger, the overall anatomy does not change).

*2) Figure 2 panels C-I: Concerning discontinuous Z lines: for a non-specialist, it is not clear from the EM what is oblique relative to what (in some panels the Z lines seem virtually parallel to other striation, presumably the filaments) and what is the relationship with the schematics in Figure 1. Maybe a higher magnification would help?*

We have added an interpretative schematic of the electron micrographs of obliquely striated longitudinal and oblique myocytes shown in Figure 2, as well as a zoom on oblique striation itself (and a schematic of it) in panel 2C.

We hope the new panels make it clearer that the (discontinuous) Z-lines (in black) serve as an anchor band to the arrays of myofilaments (in red; in cases where the anchoring point is located outside of the plane of section, it is represented by a dotted line). Regarding the definition of oblique striation: A muscle is called “cross-striated” if its myofilaments are perpendicular to the Z-lines, and “oblique” if the angle between myofilaments and Z-lines is other than 90°. In the case of the *Platynereis* obliquely striated muscles, we agree with the referees that the myofilaments are “virtually parallel” to the Z-lines, as they form a 5° angle (Figure 2) – so that the myofilaments are indeed almost (but not quite) parallel to the Z-lines.

*Contrary to what the legend says, H is not an enlargement of the yellow box in G (even if it looks similar).*

We agree. H and G actually show two consecutive cross-sections of the stomodeum of the same individual, observed at two different magnifications. Additionally, we noticed that the yellow box was originally and erroneously positioned on the contralateral body side to the one H illustrates. We have replaced the box on the correct body side and now state in the figure legend that H is another cross-section of the muscle in the yellow box, observed at a higher magnification, rather than an enlargement.

*3) Figure 3, panel B: smooth muscle MHC is expressed in many more places than the small posterior triangle at 72hpf: this should be commented on.*

We have now added comments on the other expression sites: the stomodeum (both stomodeal mesoderm and ectoderm) and internal cells of the parapodia. While a full characterization of the latter cells is beyond the scope of our paper, our preliminary observations suggest they might be part of the nephridial complex (nephridial tubule and nephridiopore – which opens at the base of the parapodium in polychaetes), which is now mentioned in the text.

*Figure 3—figure supplement 3, panel J: Why is actin considered a muscle marker? Is it smooth muscle actin?*

Phylogeny of the *actin* gene shows that the ancestral bilaterian had a single *actin* gene, which gave rise by duplication to vertebrate non-muscle, smooth muscle and striated muscle *actins* (the muscle and non-muscle isoforms of *Drosophila* arose by independent duplications – see e.g. Oota and Saitou, Mol. Biol. Evol., 1999). The *Platynereis* genome appears to contain a single *actin* gene expressed in both myocytes (at a high level – as the strongest signal is observed in the musculature in both in situ hybridizations and phalloidin stainings) and in all cells at a much lower, housekeeping level. This difference in expression level allows to use it as a muscle marker gene.

*The Panels J and K look like featureless clouds of fluorescent color that could easily be made to overlap more or less by adjusting the intensity and are not very convincing.*

We agree. Double in situ hybridizations are straightforward at early larval stages of *Platynereis* (such as 48 hpf; e.g., Denes et al., Cell, 2007) but more difficult at 72 hpf, which is the stage of earliest detectable gene expression in future smooth myocytes (at even later stages, double WMISH become impossible). The problem is compounded by the fact that visceral myocytes are located far from the surface of the animal, which represents a challenge for both permeabilization during the protocol and for imaging, and that the forming gut is prone to background staining formation (as it has endogenous alkaline phosphatase activity). For all of these reasons, the double WMISH are difficult to interpret for observers not closely familiar with *Platynereis* musculature. In spite of our best efforts, we could not optimize the double in situ hybridization protocol beyond the quality shown in the first submitted version of the manuscript. We have thus decided to remove this dataset, which was not essential to our conclusion – since we also documented smooth muscle gene expression by confocal imaging (of single WMISH) with cellular resolution at 6 dpf (Figure 3—figure supplement 4). We have also removed the brightfield panels of *foxF* and *NK3* expression at 3 days, as these were only informative in the context of the double WMISH showing the expression sites to be forming muscles. Expression of *foxF* in the visceral musculature remains supported by confocal imaging of single WMISH at 6 dpf.

*In general, Figure 3—figure supplement 3 in its current form is probably the least satisfactory piece of expression data: expression of Gata456 in the gut region is not clear at 72hpf, (if anything, on the wholemount, more of a gap than a signal in the posterior midline is what is seen) and at 6dpf it is embedded in a ubiquitous-looking expression in the posterior half of the animal, which weakens the correlation. FoxF1 and NK3 are expressed in a dotted or very patchy pattern hard to relate to MRLC and calponin. On the schematic, the red and orange should be differentiated more. These ambiguities need to be resolved.*

We agree that the panels illustrating *GATA456* expression were not clear. For this reason, we have replicated the in situ hybridizations in order to optimize the signal-to-noise ratio. These new stainings are much clearer and are now featured in our new Figure 3—figure supplement 3. We hope it is now clear that *GATA456* is far from ubiquitous, and is detected in cells covering the midgut (including midline cells – see the cells within the white outline in Figure 3—figure supplement 3)) as well as in some additional segmentally iterated bilateral cells (as also observed by confocal imaging Figure 3—figure supplement 4).

The Figure panels showing *FoxF1* and *Nk3* expression at three days have been removed (as the expression is much better documents at 6dpf). We have also changed the orange color in the schematic so that the contrast with the red color is clearer.

*4) Figure 3—figure supplement 4: Why such different shapes of the midgut given that we are looking at Z stacks of confocal views: we would expect that a more homogenous morphology could be achieved from animal to animal. Without the dashed line it would be difficult to conclude that we are looking at the same structure.*

We agree that the morphology of the midgut at this stage is variable. These differences are due to several factors: at 6 dpf, development is not perfectly synchronous anymore; the shape and position of the large lipid droplets (around which the gut forms by cellularization of the macromeres) differ between individuals (see e.g. Figure 3—figure supplement 3); and muscle contraction itself changes the shape of the animal at this stage – depending on which muscles are contracted during fixation (while variations are mild at 48 hpf and 72 hpf, when musculature development is still limited, they are more extensive in the very muscular 6 dpf larvae). Manual examination of confocal stacks confirms that the relative positions and connections of these cells to neighboring structures are the same, and thus that we are observing the same structure – the forming midgut.